# Incorporating hydrology into climate suitability models changes projections of malaria transmission in Africa

M. W. Smith [1✉], T. Willis[1], L. Alfieri[2], W. H. M. James [1], M. A. Trigg[3], D. Yamazaki[4], A. J. Hardy [5], B. Bisselink [2], A. De Roo [2], M. G. Macklin [6] & C. J. Thomas [6]

Continental-scale models of malaria climate suitability typically couple well-established temperature-response models with basic estimates of vector habitat availability using rainfall as a proxy. Here we show that across continental Africa, the estimated geographic range of climatic suitability for malaria transmission is more sensitive to the precipitation threshold than the thermal response curve applied. To address this problem we use downscaled daily climate predictions from seven GCMs to run a continental-scale hydrological model for a process-based representation of mosquito breeding habitat availability. A more complex pattern of malaria suitability emerges as water is routed through drainage networks and river corridors serve as year-round transmission foci. The estimated hydro-climatically suitable area for stable malaria transmission is smaller than previous models suggest and shows only a very small increase in state-of-the-art future climate scenarios. However, bigger geographical shifts are observed than with most rainfall threshold models and the pattern of that shift is very different when using a hydrological model to estimate surface water availability for vector breeding.

---

[1] School of Geography and Water@Leeds, University of Leeds, Leeds, UK. [2] European Commission, Joint Research Centre, Ispra, Italy. [3] School of Civil Engineering and Water@Leeds, University of Leeds, Leeds, UK. [4] Institute of Industrial Science, The University of Tokyo, Tokyo, Japan. [5] Department of Geography and Earth Sciences, Aberystwyth University, Aberystwyth, UK. [6] School of Geography and Lincoln Centre for Water and Planetary Health, University of Lincoln, Lincoln, UK. ✉email: m.w.smith@leeds.ac.uk

Malaria is a climate-sensitive vector-borne disease that was responsible for an estimated 435,000 deaths from 219 million malaria cases worldwide in 2017; 92% of these malaria deaths were reported in Africa[1]. Detailed mapping of current malaria transmission is vital for the distribution of health resources and targeting of control measures. Moreover, an understanding of the environmental conditions required for malaria transmission is necessary for predicting areas vulnerable to future outbreaks. Future climate change is likely to affect the distribution and intensity of malaria transmission, though the exact nature and extent of this influence has been the subject of recent debate[2].

Ambient air temperature controls the rate of several components of the malaria transmission cycle including sporogonic and gonotrophic development rates, biting rate and individual longevity[3]. Extensive laboratory and field studies have led to a greater understanding of the suitable temperature ranges of both parasite and vectors[4,5], and the effect of water temperature on larval development[6,7]. Although uncertainties remain with regard to the temperature parameters of malaria climate suitability models[8–10], recent results[8] suggest a nonlinear unimodal temperature-response model between 16 and 34 °C is appropriate. In any case, temperatures across much of Africa are suitable for malaria transmission for such temperature suitability curves[11].

The availability of water at the ground surface for vector mosquito larval habitats is another critical environmental control of malaria transmission[12–14]. Estimation of surface water availability from current global datasets is challenging. Instead, monthly rainfall is typically used as a proxy for habitat availability; thresholds of 60 or 80 mm month$^{-1}$ have gained traction as a proxy for breeding habitat in Africa[15–17] and have been applied across the globe[18]. However, since complex and spatially variable hydrological processes (e.g. infiltration, evaporation, soil moisture storage, transfer through and storage in river networks) are omitted, a wide variety of rainfall thresholds are found in the literature[19–24] that leads to large differences in environmental suitability estimates. Moreover, irrigated areas and reservoirs arising from the construction of large dams have been observed to provide suitable year-round habitat for Anopheles mosquitoes[25,26] but are not included in such models. Although detailed dynamic process-based models coupling hydrological and biological components of malaria transmission have been developed at the village-scale[13,14], such hydrological representations are lacking in existing malaria climatic suitability estimates at the continental scale.

Here we use daily climatic projections from seven downscaled general circulation models (GCMs) to estimate current and projected changes in the number of months hydro-climatically suitable for malaria transmission in Africa over the next century. We demonstrate the sensitivity of present and future estimates of hydro-climatic suitability to the representation of water body availability and show that the application of a hydrological model to make these estimates produces a more realistic, though more complex pattern of malaria transmission that has considerable implications for present estimates and future predictions of populations in stable and unstable malaria transmission zones.

## Results

**A more complex pattern of malaria climatic suitability.** The three thermal response curves[8–10] vary little terms of the area estimated to be suitable for malaria transmission with a maximum difference of 2.86 Mn km$^2$ or 9% of Africa total surface area (Supplementary Fig. 1). Conversely, the range of area estimates using the different rainfall thresholds proposed is much greater (16.14 Mn km$^2$, 53% of Africa total surface area) reflecting the wide

variability of thresholds chosen to represent local hydrological conditions (Fig. 1).

We couple the Lisflood hydrological model with a model of malaria climate suitability. This LIS-MAL model (Supplementary Fig. 2) produces a more realistic pattern of hydro-climatic suitability than that based on a simple rainfall threshold (Fig. 2a, b and Supplementary Fig. 4). Herein, we primarily compare the LIS-MAL hydrological model estimates with a moderate rainfall threshold of 60 mm month$^{-1}$ alongside single catalyst month of 80 mm rainfall as applied by Tanser et al.[22]; comparison with other published rainfall thresholds is presented in the Supplementary Information. Although flowing water in large river channels does not provide suitable larval habitat for African vector mosquitoes, associated smaller water bodies in adjacent bankside and floodplain areas can be highly productive. Large-scale river networks are clearly identifiable as potential foci for year-round malaria transmission when a hydrological model is used to represent water availability. In particular, the Nile system extends prominently to the north coast of Africa. Although year-round transmission may be an overestimate, the extension of suitability this far north is supported in part by historical observations of malaria outbreaks along the Nile corridor[27–29] (Fig. 2d and Supplementary Information S2). The Niger and Senegal rivers in Mali and Senegal, and Webi Juba and Webi Shabeelie rivers in Somalia similarly extend beyond the geographical ranges predicted to be climatically suitable for malaria by all rainfall thresholds but are observed foci of malaria transmission in national-scale surveys[30,31].

Quantitative validation of estimates of hydro-climatic suitability for malaria is problematic owing to malaria transmission being driven by more than climate alone. We present both quantitative and qualitative validation against observations in Supplementary Notes 2. Although there have been substantial and spatially variable changes in temperature and rainfall over Africa since 1900[32], the pre-intervention map of Lysenko and Semashko[27] (Fig. 2d) is most appropriate for comparison. Overprediction of malaria suitability in South Africa is less extensive than with the rainfall threshold (Supplementary Fig. 5) while under-prediction in the Horn of Africa region likely relates to pronounced drying trends in the twentieth century[33]. LIS-MAL appears to underestimate the extent of malaria suitability in east Africa shown on the pre-intervention map; however, we note that these areas in Kenya and Tanzania are described[2] as 'malaria near water', so the mismatch may well be due to the broad terminology used by Lysenko and Semashko[27]. Overall, hydro-climatic models perform similarly across the range of validation layers presented. Although there is no evidence to favour any single model, intermediate rainfall thresholds (such as that of Tanser[22]), the rainfall to evapotranspiration ratio of Lindsay[24] and the more detailed hydrological treatment of LIS-MAL strike a balance between the more extreme rainfall thresholds.

As water is routed through the landscape in Lisflood, a much larger area is observed to be hydro-climatically suitable for year-round malaria transmission than predicted using the rainfall threshold (Fig. 2e). Adding mapped irrigated areas to the LIS-MAL model further increases both the area and season length hydro-climatically-suitable for transmission (Fig. 2c). The area estimated to be climatically suitable for malaria transmission for >3 months is smaller when Lisflood is used; yet, the differences in estimates of populations in hydro-climatically suitable areas for stable malaria transmission are less pronounced owing to LIS-MAL identifying waterways of high population density that are foci for year-round transmission (Fig. 2f and Supplementary Table 8).

**Different future patterns of malaria suitability.** Predictions of future climatic suitability for malaria transmission averaged over all

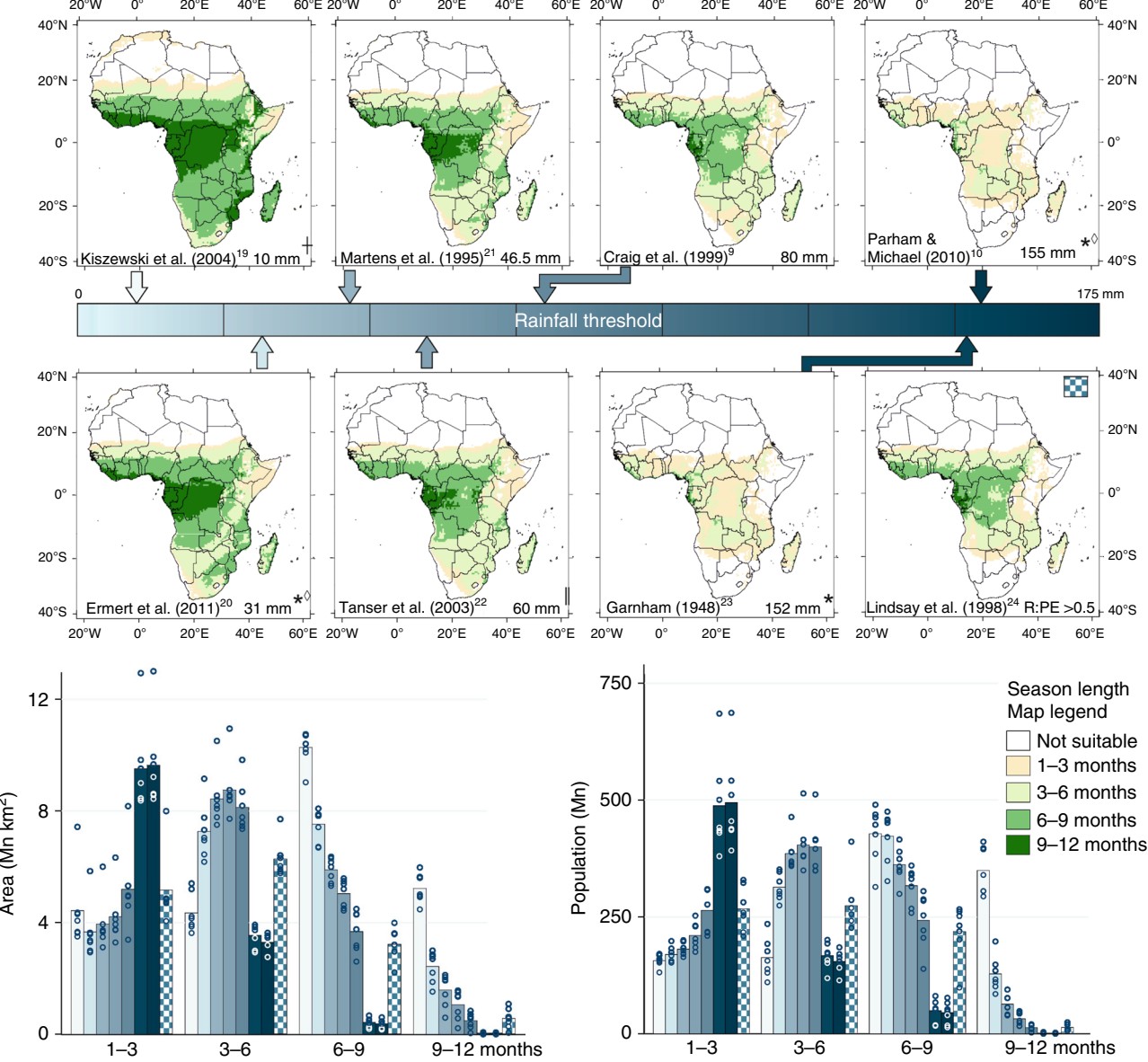

**Fig. 1 The effect of rainfall threshold on estimates of climatic suitability for malaria transmission.** The estimates from eight rainfall thresholds reported in the literature (see Supplementary Table 1 for further details) combined with the Mordecai et al.[8]. thermal response curve for Africa over the period 1971–2005. Areas within 3-month bands are calculated for season length, with the coloured arrows locating each rainfall threshold on the linear scale bar providing a legend. Bars indicate mean of model estimates ($n = 7$); points indicate individual model estimates. *Originally calculated for daily (or dekadal) rainfall and scaled up for a monthly value; || includes a 'catalyst month' of 80 mm rainfall; ◊ also implements an upper rainfall threshold (Supplementary Table 1); † only for 'temporary' water bodies that form the breeding habitat of the *Anopheles gambiae s.l.* complex.

seven GCMs (under the Representative Concentration Pathway 8.5 scenario) vary substantially with the representation of hydrological processes (Fig. 3a) and model agreement generally declines slightly through time (Supplementary Table 4). In the majority of models, an increase in suitable area is preceded by a decrease from 1971–2005 to 2011–2040, providing important context to recent observed changes in malaria transmission (Supplementary Table 5). Much of the subsequent increase is realised by 2041–2070. Similar patterns emerge considering the areas suitable for stable transmission (>3 continuous months); most models predict an increase in area by 2071–2100, though both LIS-MAL and Tanser[22] rainfall threshold models suggest this increase will be very small (0.08 and 0.03 Mn km², respectively). More pronounced increases in malaria suitability predicted using alternative rainfall thresholds may, therefore, be unfounded.

The predicted future change in total area hydro-climatically suitable for stable malaria is more sensitive to the choice of thermal model than hydrological representation owing to the stronger directionality of future temperature changes. However, the shift in the location of these areas is similarly sensitive to both hydrological and temperature representation (Fig. 3, Supplementary Figs. 6–8 and Supplementary Tables 6, 7). Nearly all hydrological representations predict decreasing duration of malaria suitability either in West Africa, southern Africa or in both locations; however, this is dominated by areas experiencing small average changes of <1 month. Multiplying the area by the number of months change in malaria hydro-climatic suitability provides a clearer indication of the magnitude of the changes. By this 'month-area' metric, the hydrological representations provide varied predictions with many (including the Tanser rainfall

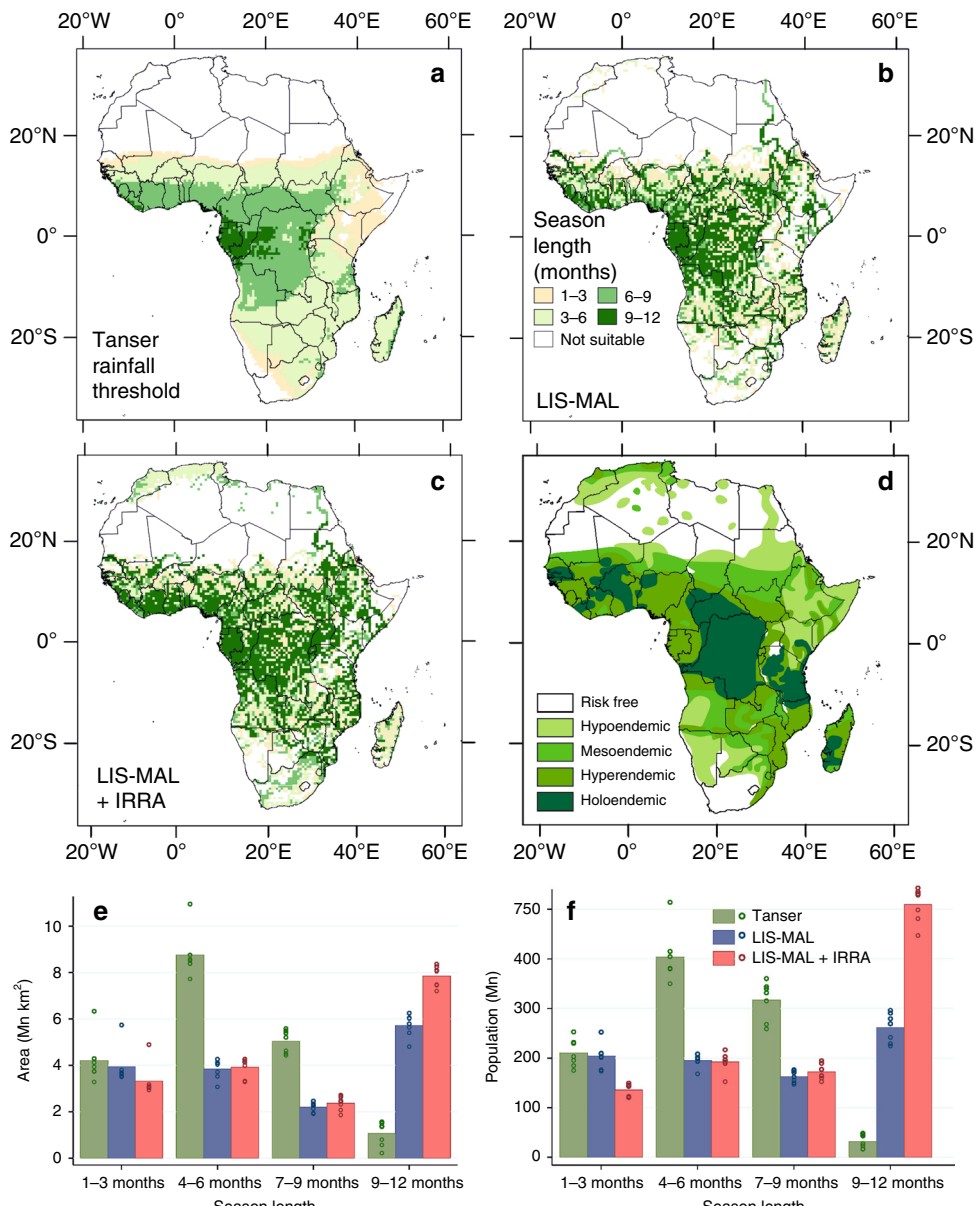

**Fig. 2 Comparison of estimates of climatic suitability for malaria transmission in Africa.** Results from the LIS-MAL model over the period 1971–2005 are compared with those using a 60 mm month$^{-1}$ rainfall threshold (with an 80 mm catalyst month)[22]. Thermal response in both is represented using the Mordecai et al.[8]. curve for consistency. Mean values of climatic projections from each of seven downscaled general circulation models (detailed in Supplementary Table 2) for: **a** the Tanser rainfall threshold; **b** LIS-MAL; and **c** LIS-MAL with mapped irrigation compared with **d** the pre-intervention malaria map of Lysenko and Semashko[27]. Comparison of estimated area (**e**) and population (**f**) in 3-month bands of (hydro-) climatic suitability for malaria transmission. Bars indicate mean of model estimates ($n = 7$); points indicate individual model estimates.

threshold model) showing an initial increase in malaria-months followed by a decrease. LIS-MAL shows a relatively stable net value with increases offset by decreases; however, the magnitude of the shift in geographic area is relatively large compared with other models.

Differences are observed in the location, extent and severity of areas predicted to experience a change in hydro-climatic malaria suitability. Both LIS-MAL and the Tanser rainfall threshold model predict an increase in malaria in the Ethiopian Highlands, driven by temperature changes (Fig. 3b). Increases in hydro-climatic suitability are also observed around East Africa where a warmer and wetter climate is predicted with hydrological changes being the main driver (Fig. 3b and Supplementary Figs. 9, 10). Increased malaria hydro-climatic suitability in South Africa and

Lesotho (where warming is accompanied with a projected increase in aridity) exhibits distinctly different patterns: rather than being focused in the east of the country centred on Lesotho as in the rainfall threshold model, LIS-MAL predicts the area of increased suitability to stretch along the course of the Caledon and Orange rivers to the border with Namibia owing to continued water availability along the river corridors. The expanding wedge of hydro-climatic suitability across Niger predicted by the rainfall threshold is not reproduced by LIS-MAL for which projected increases in suitability are more pronounced further south around the Gulf of Guinea. The predicted widespread aridity-driven decrease in hydro-climatic suitability across southern Africa especially in Mozambique and Botswana by the Tanser rainfall threshold is not observed in LIS-MAL where water availability is

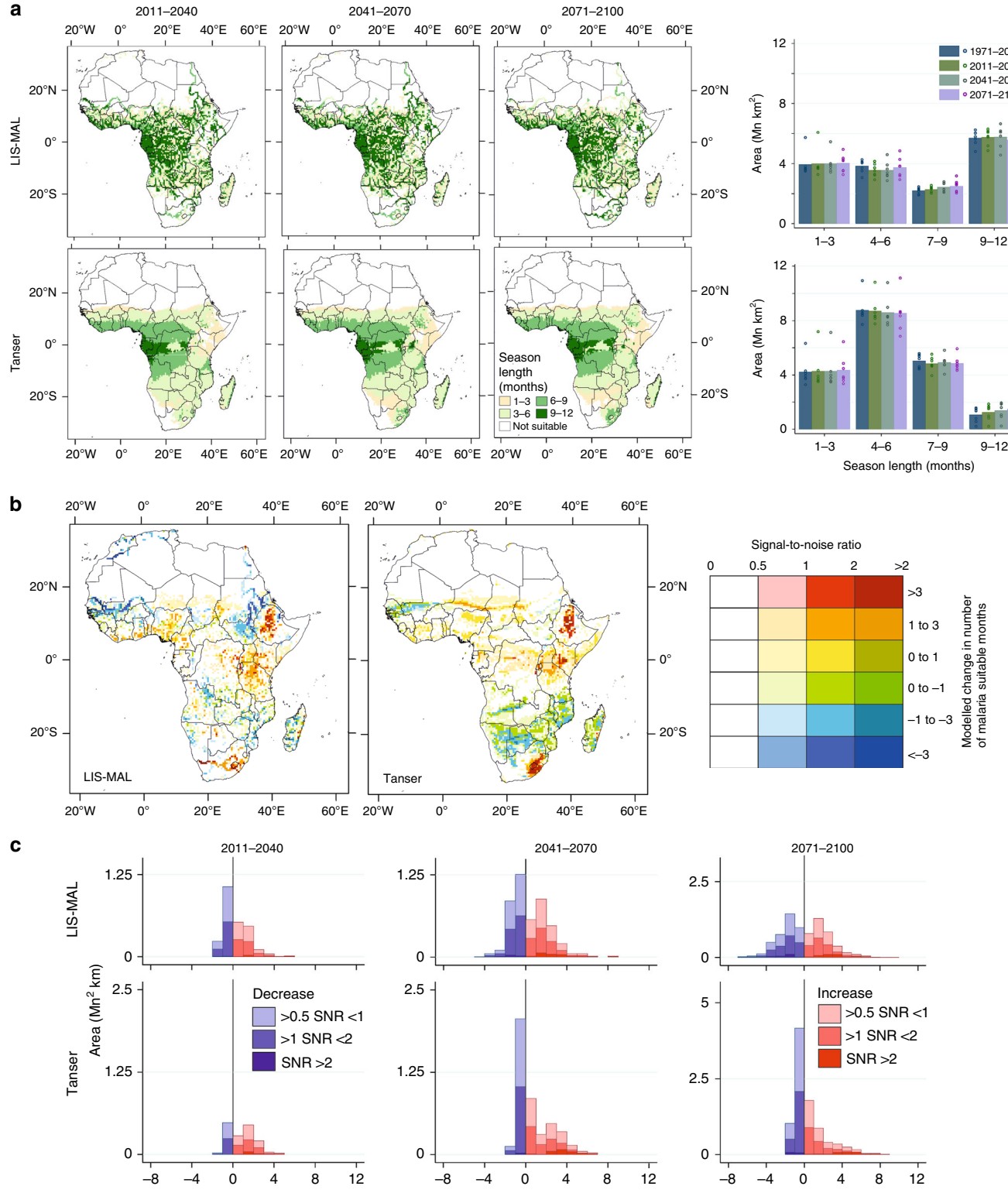

**Fig. 3 Future predictions of climatic suitability for malaria transmission. a** Mean model predictions for LIS-MAL and the Tanser rainfall threshold at each time step with a summary of areas in 3-month categories. Bars indicate mean of model estimates ($n = 7$); points indicate individual model estimates. **b** Mean changes in malaria hydro-climatic suitability predicted across forcing models by LIS-MAL and Tanser models between 1971–2005 and 2071–2100, where the signal-to-noise ratio >0.5. Different saturations indicate the signal-to-noise ratio across the seven projections, with the noise defined as the standard deviation of estimates across the projections. **c** Stacked histograms showing changes number of months climatically suitable up to 2071–2100 split into categories of signal-to-noise ratio from **b**. For changes between all periods and all hydrological models see Supplementary Figs. 6–8.

projected to remain sufficient to maintain transmission. In agreement with a previous, more detailed study in West Africa[14] both models predict a decrease in suitability in Mali and further west driven by increased aridity and encroachment of the maximum thermal limit for malaria, but this decrease is more widespread in LIS-MAL predictions. Finally, the most pronounced difference between the two models is the substantial reduction in malaria suitability in South Sudan predicted by LIS-MAL, which stretches all along the Nile system. In this case, decreases in both thermal and hydrological suitability coincide and the pronounced effect on estimated malaria transmission suitability is amplified by the temperature-dependent larval development rate in the estimation of hydrological suitability in the LIS-MAL model (Supplementary Figs. 2 and 11).

The differences in population estimates between the two malaria models are magnified in future projections with the Tanser rainfall threshold predicting larger numbers affected (Fig. 4 and Supplementary Table 9). In this high concentration scenario (i.e. RCP 8.5), both models predict an increase in populations living in areas suitable for stable malaria transmission with an additional 2238 Mn (Tanser rainfall threshold) and 1765 Mn (LIS-MAL) by 2071–2100; this is driven by projected population increases rather than changing areas as numbers affected can increase despite a modelled decrease in hydro-climatic suitability. The choice of UN population variant (Supplementary Fig. 12) has a large effect on these values, though the overall pattern remains the same. The Tanser rainfall threshold model estimates greater numbers affected in southern Africa and parts of West Africa (Supplementary Fig. 13) and larger percentage increases in affected population around Chad, Niger and Nigeria in central Africa and also in Angola. Percentage increases in population-months estimated by LIS-MAL are more focused around east Africa, particularly Somalia and Tanzania.

## Discussion

Malaria is a complex disease and changes in transmission cannot be attributed to climate alone[34,35]. Here we restrict our analysis to quantifying changes in areas that are considered to be hydro-climatically suitable for malaria transmission and the sensitivities of these estimates to the assumptions employed. We have made an important initial step in employing established hydrological models to make process-based predictions of hydro-climatic malaria suitability as opposed to using simple rainfall thresholds. This includes the potential for future high-resolution models: to quantify water body dynamics at catchment and floodplain scales a hydrodynamic river routing model[36] could be used to represent flood expansion and contraction at sub-km spatial resolutions. This would also allow a more complex environmental estimate of malaria hydro-climatic suitability by enabling the modelling of finer-scale hydrological processes determining the availability of Anopheles breeding habitat, permitting other physical processes (e.g. larval flushing effects of high-velocity river flows) to be included and for vector niches to be represented explicitly. Such an approach would help bridge the scale gap between continental and landscape-scale malaria environmental risk estimates, so that projections of climate change impacts can be interpreted at the operational scale of public health interventions.

## Methods

**Climate projections**. We used a set of seven climate projections with a high concentration scenario (i.e. RCP 8.5) produced with EC-EARTH3-HR v3.1[37] by the Swedish Meteorological and Hydrological Institute as used previously by Alfieri et al.[38]. Forcing data are derived from seven independent driving GCMs produced within the Coupled Model Intercomparison Project Phase 5 (listed in Supplementary Table 2). Downscaled projections are obtained by forcing EC-EARTH3-HR with sea surface temperature and sea-ice concentration from each of the seven

GCMs as boundary conditions, yet preserving the original global extent. Model outputs are downscaled from their individual grids to a common spatial resolution of 0.35° (~40 km at the equator). No bias correction was performed on the climate projections as it often breaks the physical coherence between the atmospheric variables, whereas its benefits in regions with sparse observational datasets as in this case are yet disputed[39]. Resulting runoff simulations were shown to have variable quality, with positive bias in arid regions, yet skillful in reproducing their sub-seasonal variability[40].

**Hydrological modelling**. The Lisflood hydrological model[41,42] was set up at 0.5° resolution (~55 km at the equator) and forced using daily temperature and precipitation data from the seven climatic projections over a period of 130 years, starting in 1971. Daily potential evapotranspiration was also calculated from daily mean temperature, wind speed, relative humidity and solar radiation using the Penman–Monteith equation. Lisflood calculates a complete water balance at a daily time step and every grid cell defined in the model domain. Snowmelt, soil freezing, surface runoff, infiltration into the soil, preferential flow, redistribution of soil moisture within the three-layer soil profile, drainage of water to the groundwater system, groundwater storage and groundwater base flow are all simulated for each grid cell. Runoff produced for every grid cell is routed through the river network, using a kinematic wave approach. Lakes, reservoirs and retention areas are simulated by giving their location, size, inflow and outflow boundary conditions. Lisflood also requires distributed elevation data (derived from the Hydrosheds database[43]), channel geometry (taken from the work of Wu et al.[44]), soil texture and depth (derived from the ISRIC 1 km SoilGrids database[45]) and land-use (monthly climatic mean Leaf Area Index maps were derived from SPOT-VGT data and were assumed constant for future projections) and was set up as per Alfieri et al.[38].

**Estimation of malaria hydrological suitability**. Daily mean surface air temperature, rainfall, and runoff data were calculated and divided into four time periods: 1971–2005, 2011–2040, 2041–2070 and 2071–2100. Average daily temperature, potential evapotranspiration and total rainfall in each month of the year were averaged for each time period. All data were transformed into an equal area projection (Africa Albers Equal Area Conic) appropriate for area calculation over the African continent. From Lisflood daily runoff data a minimum threshold of $30\ m^3\ s^{-1}$ (equivalent to 1 mm depth across the grid cell) was established as a threshold for suitability, indicating sufficient water availability for a given grid cell. A sensitivity analysis for this threshold is presented in Supplementary Information S1: overall, LIS-MAL suitability estimates were not sensitive to the runoff threshold for values ±50% (increasing and decreasing the threshold value by 50% changed the estimated suitable area by −6% and 10%, respectively). The number of days this condition is met per month was then averaged over each time period; when this exceeded the temperature-dependent development period of Anopheles mosquitoes as estimated by the model of Bayoh and Lindsay[6], the month was considered to be hydrologically suitable. The average daily mean surface air temperature was used to establish the development rate to maturity, though where this is below the 31 °C optimum, we apply a +2 °C offset as suggested by Paaijmans et al.[46] to account for the typically higher water temperature. Since shading is often available, we do not apply this offset for temperatures above the optimum as larvae are likely to seek optimum conditions where available.

**Irrigation**. Irrigated areas are included separately using the IWMI Irrigated Area map of Africa (2010) (http://waterdata.iwmi.org/applications/irri_area/), which identifies irrigated croplands at 230 m spatial resolution and excludes areas of rainfed agriculture. Irrigated areas are considered to contain suitable water bodies year-round[25]. To incorporate the flight range of the mosquito vector, we apply a 3 km buffer around these features and consider this area hydrologically suitable as surface water is available within the mosquito flight range. The 3 km buffer follows the findings of Thomas et al.[47] that 95% of A. gambiae populations in rural savannah areas of The Gambia did not disperse beyond this distance from the breeding site. This buffer was not applied to other hydrological layers given their much larger grid-cell length, which implicitly includes flight ranges. Modelled runoff and mapped irrigation are combined to produce a monthly evaluation of hydrological suitability for malaria transmission. This combination was not used for future projections given the synoptically mapped irrigation component.

**Hydro-climatic suitability and comparison with rainfall thresholds**. To enable direct comparison between rainfall thresholds from the literature, we have converted all stated thresholds to monthly values; full details of the thresholds as originally applied are reported in Supplementary Table 1. We include any upper thresholds or 'catalyst months' within our comparison and note that the threshold of Kiszewski et al.[19] is intended to represent only 'temporary' water bodies; however, given that these are the preferred habitat of the Anopheles gambiae s.l. complex, this threshold is important for our modelling. To compare precipitation thresholds with hydrological model outputs for predictions of hydro-climatic suitability of malaria transmission, we apply the most recent temperature curve of Mordecai et al.[8] to assess changes in season length. Our focus is on the range of hydro-climatic suitability and thus the full range of viable temperatures (16–34 °C)

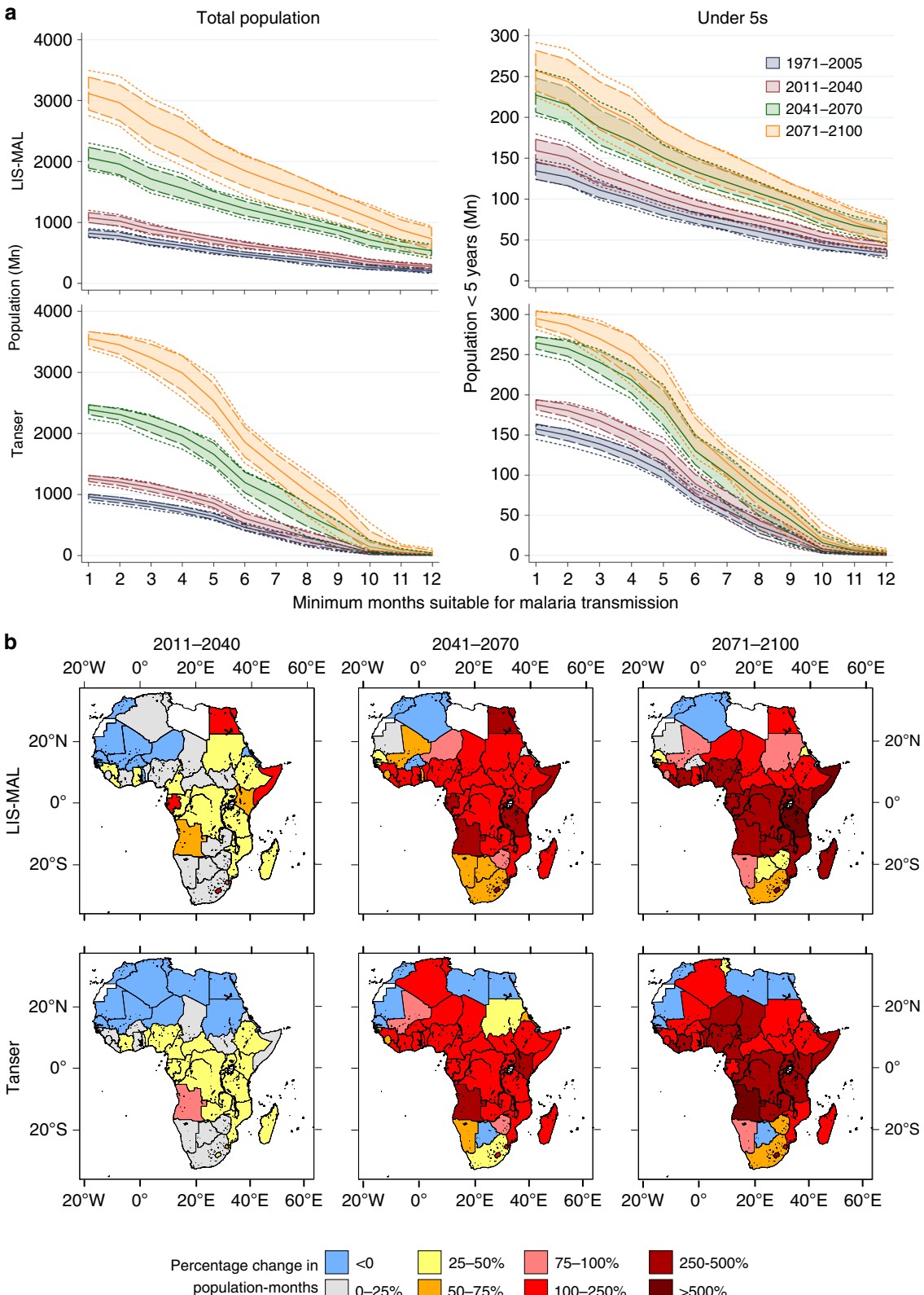

**Fig. 4 Future populations in areas climatically suitable for malaria transmission. a** Populations within areas climatically suitable for malaria transmission. Solid lines show the mean of the GCM forcing models, shaded areas indicate ±1 standard deviation and dashed lines show a range of predictions. Total population and under-5s calculated for both the Tanser rainfall threshold and LIS-MAL estimates using the WorldPop gridded distributions and UN medium variant projections for each period. **b** Predicted percentage changes in population-months (i.e. population exposed multiplied by the number of additional months of exposure) by country. White shading indicates no estimated malaria suitability.

is considered suitable. The Mordecai et al.[8] temperature ranges were combined with hydrological suitability layers to create a monthly mask of areas hydro-climatically suitable for malaria transmission. To illustrate the differences in modelled malaria transmission when using our Lisflood hydrological model versus a rainfall threshold approach, we consider the widely used 60 mm month$^{-1}$ rainfall threshold (with the additional requirement of a catalyst month of 80 mm rainfall) as detailed in Tanser et al.[22]. Comparisons with other thresholds are provided in the Supplementary Information. Season length was determined as the maximum number of continuous months of hydro-climatic suitability. Subsequent data analysis was undertaken in Stata 12.1.

**Temperature suitability curves**. To compare the effect of rainfall thresholds on estimated malaria suitability with that of the temperature ranges applied, we also compare the temperature curve of Mordecai et al.[8] with those of Craig et al.[9] and Parham & Michael[10], again considering the full temperature range as suitable for transmission. In addition to the minimum (18 °C) and maximum (40 °C) mean monthly temperature of Craig et al.[9], we apply their frost criterion to exclude any area with a minimum monthly temperature of <4 °C. For direct comparison, we apply the Parham & Michael[10] temperature suitability curve to the same monthly data and again use the full range of suitability, between a minimum of 20 °C and a maximum of 40 °C. We calculate and compare the land area within the suitable malaria transmission temperature range for each of the three models for each month of the year. We then apply the 80 mm month$^{-1}$ rainfall threshold[9] to compare predicted areas of climatic malaria suitability, defined as the maximum number of continuously suitable months (i.e. season length).

**Population estimates**. Gridded estimates of the human population in Africa were used to evaluate the number of individuals falling within areas climatically suitable for malaria transmission. The analysis was performed for the total population[48] and for children aged under 5 years of age[49] as this group is particularly susceptible[1,50]. All population datasets were at a resolution of 30 arc seconds (~1 km at the equator) with results aggregated to the continental and country level. For initial population estimates, we used the 2015 WorldPop grids, while for future scenarios, we rescaled the WorldPop 2020 grids to match country level UN projections (total population and under 5's, medium variant) for the mid-point of each time period[51]. For comparison, we also calculated the populations under the UN low and high variants.

**Reporting summary**. Further information on research design is available in the Nature Research Reporting Summary linked to this article.

## Data availability

LIS-MAL estimates of hydro-climatic suitability for malaria transmission in Africa (1971–2100) can be downloaded here[52] https://doi.org/10.5518/786. Source data are provided with this paper. Data used are available here: https://data.jrc.ec.europa.eu/collection/floods. SPOT-VGT data are available at https://land.copernicus.eu/global/products/lai, Hydrosheds elevation data at https://www.hydrosheds.org/downloads and the SoilGrids Database at https://www.isric.org/explore/soilgrids. Source data are provided with this paper.

## Code availability

The LISFLOOD model code is available for download here https://github.com/ec-jrc/lisflood-code. Python scripts for extracting monthly hydro-climatic variables from daily NetCDF climate data are provided here[52] https://doi.org/10.5518/786.

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

## Acknowledgements

This work was undertaken with support from the UK Natural Environment Research Council awards HYDROMAL (NE/H022740/1) and FLOODMAL (NE/P013481/1) to C.J.T., M.W.S., A.J.H. and M.G.M.

## Author contributions

M.W.S. and C.J.T. were involved with the conceptualisation of this paper. L.A., B.B. and A.D.R. ran the Lisflood model simulations. M.W.S. and T.W. conducted modelling of malaria suitability. W.H.M.J. performed the population analysis. M.W.S. and C.J.T. were responsible for original drafting and preparation. M.G.M., A.J.H., M.A.T. and D.Y. reviewed and substantively edited content. M.W.S. prepared the final manuscript and figures.

## Competing interests

The authors declare no competing interests.
