## [Peer Review File · Nature Communications]

Reviewers' Comments:

Reviewer #1:

Remarks to the Author:

Review of "Hydro-climatic suitability of malaria transmission in Africa: new patterns emerge"

Manuscript number: NCOMMS-20-04720-T

Authors: Smith et al., 2020

Recommendation: Major revisions

General:

This study employs a novel malaria model that combines irrigation effects, temperature suitability curves and the LisFlood hydrological model in order to derive high resolution maps of malaria risk in Africa. Future projections are provided to investigate climate change effects. There is an excellent section comparing different published rainfall models and thresholds, and novel estimates of population at risk. Overall the paper is well written, the analysis and plots are neat and clear. The approach is quite novel and should be published in Nature communications. However, I have several major comments that need to be addressed before acceptance of the manuscript. I encourage the authors to address my comments.

Major points:

1] Modern estimate of malaria risk to add

The authors provide pre-control map of malaria endemicity risk by Lysenko et al. As the model is driven by climate parameters it makes sense. I think the authors should provide recent risk maps of *P. falciparum* infections in Africa would be a plus. The authors could use simulated prevalence data from the Malaria Atlas project (data can be downloaded in the explorer - <https://malariaatlas.org/>) to discuss the recent context (and discrepancies with climate driven malaria models).

2] Large malaria season values simulated over the Sahelian fringe

One of my main concern, the Lisflood and Lisflood/irrigation model seems to overestimate the simulated length of the transmission season. In Senegal for example, clinical malaria cases tend to be reported in Oct-Nov-Dec (3 months roughly which fits the old Tanser model) with very few cases during the dry season. See the work by I. Diouf for example (<https://doi.org/10.3390/ijerph14101119>). Year-round transmission values in the Nile region must be backed up by medical papers / epidemiological studies as well. This overestimation appears clearly on Fig 2e-f where LisFlood predicts a very large surface suitable for 9-12 months. DRC values should be larger I think, this is one of the most endemic country in Africa. Basically, the authors should provide more details on these issues in discussion.

3] Climate Model bias-correction?

A few points here. Firstly, the authors should emphasize that they are using a Regional Climate Model driven by an ensemble of 7 GCMs in the abstract. Secondly, the authors should use a bias-calibration technique. RCM driven by GCM can show large temperature and rainfall biases over Africa. Consequently, this point needs to be addressed or at least mentioned as an important caveat in discussion.

4] Malaria model description to be improved.

First, I think the authors should find an acronym for their malaria model (as Lisflood is already a hydrological model?). The main model description in methods should be extended in Supp Materials. A diagram or schematic showing the different input layers (irrigation, population etc) and if they vary in space and/or time would be a good addition, or this information can also be included in a summary table.

5] Other hydrological malaria models could be mentioned in Introduction

I am thinking about the work by Arne Bomblies & Yamana et al.

(<https://doi.org/10.1029/2008WR006917> -

<https://malariajournal.biomedcentral.com/articles/10.1186/1475-2875-8-223> -

<https://www.nature.com/articles/nclimate3085>). These other models developed for small scale applications should be compared to the main author's findings as well.

Minor points:

Intro: "...malaria deaths were reported in Africa."

Intro: "... biting rates and individual longevity (add references)"

Intro: "of both parasite and vectors". Importantly the authors should investigate the recent work by Waite et al (<https://doi.org/10.1098/rsbl.2019.0275>) who showed lower temperature thresholds for sporogony in key Anopheles vectors.

Intro: "The availability ... of malaria transmission (add references to Bomblies and Yamana work)"

Intro: "Moreover... not included in such models". The authors could add a couple of sentences about potential dam effects on malaria, it's a very important issue (see work by Solomon Kibret - <https://malariajournal.biomedcentral.com/articles/10.1186/s12936-015-0873-2>).

Page 7: "Lisflood projected percentage ... Tanzania". This sentence is poorly written, please reword

Conclusion: "... could be attributed to climate alone". The work by Gething et al. should also be cited here (<https://www.nature.com/articles/nature09098>)

Reviewer #2:

Remarks to the Author:

Hydro-climatic suitability of malaria transmission in Africa: new patterns emerge

This is a well-written paper that develops the science on malaria hazard. The authors show that the use of a hydrological model to map climatic suitability for malaria transmission produces a better match with the pre-intervention malaria map of Lysenko and Semashko. This is a nice result and provides confidence in their subsequent predictions of future climatic suitability.

I have been asked to comment on the hydrological modelling specifically. I believe LISFLOOD is a good choice of model for this application, and my only slight query is with the use of the 30m3s-1 minimum threshold for suitability. In particular, the authors state that the area estimated to be hydro-climatically suitable for stable malaria transmission is

smaller than previous models suggest. I wonder if this result stands if a different threshold is used.

Firstly, for clarity, I would recommend that the authors cite the source of this threshold (Kiszewski et al.(2004)) within the main body of text (rather than just in the supplementary information), secondly, I would like to see a sensitivity analysis of this threshold (not necessarily in the main body of text, but definitely in the supplementary material). There is always a challenge in relating the real world to the model world, especially to models with a coarse spatial representation of hydrology. The suitability threshold should be seen as an effective parameter, in that it is used to attempt to represent suitable accumulations of water far smaller than the grid scale. Is the area still smaller than previous models suggest if different values for this suitability threshold are used?

Liz Stephens

Reviewer #3:

Remarks to the Author:

This study applies a continental-scale hydrologic model combined with malaria temperature suitability to produce a mapping of current and future climate suitability for malaria transmission. In theory, the addition of hydrological processes and irrigation should be an improvement over models that use rainfall alone as they would provide a more accurate measure of suitable breeding habitat for the *Anopheles* mosquitoes that transmit malaria.

However, there is little validation of the model to prove that this is the case. Understandably – validation is challenging, as the authors note. The validation presented here consists of a visual comparison to the map of malaria transmission limits in 1900 constructed by Lysenko & Semashko by interpolating historical records with climate data and expert opinion. From Figure 2 of the paper, it is not obvious to me that the new model outperforms the Tanser model at reproducing the Lysenko map. From Figure S4, the model appears to do rather poorly at the northern and southern limits of malaria transmission, as well as in the East African highlands. This discrepancy is problematic, as these are precisely the areas that are most sensitive to changes in transmission potential due to climate change.

Interestingly, the hydrological model presented here is also applied as a simple threshold rule (30 m³/s over a 0.5-degree grid cell). It is unclear how this threshold relates to the fine-scale hydrological processes that determine *anopheles* habitat availability, given the species' preference for puddles and small water pools on the order of meters to tens of meters.

While the inclusion of hydrology and irrigation are useful and there are some interesting results presented as far as climate change projections, I believe the model must first be more convincing in defining malaria suitability in the present climate before we can begin to consider its ability to make projections in a new climate.

Specific comments:

1. The model needs to be better validated. Comparison with the Lysenko map should be better quantified. Additional comparisons to higher quality data, for example malaria prevalence surveys such as those used by Tanser or anopheles occurrence surveys, would greatly improve the credibility of the model. The Malaria Atlas Project provides some such data on their website.
2. The discrepancy at the northern and southern limits of climate suitability, as well as East Africa, should be addressed.
3. The choice of the rainfall runoff threshold value should be discussed, and sensitivity to this value should be assessed.
4. The discussion should mention the disconnect between the scale of 55km grid cell hydrologic modeling and finer-scale processes determining availability of anopheles breeding habitat.
5. There should be mention of the work in the decade+ following Tanser and other simple threshold models that use hydrologic processes and/or mosquito larval biology in determining climate suitability for malaria
6. This is more a matter of preference, but I believe there is too much emphasis in the manuscript on drawing comparisons between projections under Tanser's formulation vs Lisflood – I don't believe the Tanser projections are considered to be definitive or a consensus view. If a comparison to other maps or projections is necessary, there have been a number of more recent projections that may be more relevant.

REVIEWER 1

General

This study employs a novel malaria model that combines irrigation effects, temperature suitability curves and the LisFlood hydrological model in order to derive high resolution maps of malaria risk in Africa. Future projections are provided to investigate climate change effects. There is an excellent section comparing different published rainfall models and thresholds, and novel estimates of population at risk. Overall the paper is well written, the analysis and plots are neat and clear. The approach is quite novel and should be published in Nature communications. However, I have several major comments that need to be addressed before acceptance of the manuscript. I encourage the authors to address my comments.

We would like to take the opportunity to thank the reviewer for these positive comments regarding the paper. We detail our response to these areas for greater focus below.

(1) Modern estimate of malaria risk to add

The authors provide pre-control map of malaria endemicity risk by Lysenko et al. As the model is driven by climate parameters it makes sense. I think the authors should provide recent risk maps of *P. falciparum* infections in Africa would be a plus. The authors could use simulated prevalence data from the Malaria Atlas project (data can be downloaded in the explorer - <https://malariatlas.org/>) to discuss the recent context (and discrepancies with climate driven malaria models).

We have added a Supplementary Information item on validation against observations. We now validate our model against two datasets: (1) Lysenko and Semashko (1968) pre-intervention (~1900) map of malaria endemicity in Africa; and (2) Kyalo et al. (2017) anopheles inventory of sub-Saharan Africa (1898-2016) observations of anophelines (new Figure S17 and new Tables S10-11). We also investigated (3) Malaria Atlas Project modelled estimates of parasite rate from Bhatt et al. (2015) for 2015 and (4) Wiebe et al. (2017) estimate of the mean modelled relative probability of occurrence of *Anopheles gambiae*; however, these latter data sets rely in some part on gridded climate data for interpolation and cannot therefore serve for validation analysis independent of climate. We did perform the same validation on these data sets for completeness and note that there were no major differences.

We have also added sentences to the main text highlighting this extra material: *"We present both quantitative and qualitative validation against observations and modelled data in Supplementary Information S2"* and providing an overview of the results: *"Overall, hydro-climatic models perform similarly across the range of validation layers presented. While there is no evidence to favour any single model, intermediate rainfall thresholds (such as that of Tanser⁷), the rainfall to evapotranspiration ratio of Lindsey⁹ and the more detailed hydrological treatment of LIS-MAL strike a balance between the more extreme rainfall thresholds."*

(2) Large malaria season values simulated over the Sahelian fringe

One of my main concern, the Lisflood and Lisflood/irrigation model seems to overestimate the simulated length of the transmission season. In Senegal for example, clinical malaria cases tend to be reported in Oct-Nov-Dec (3 months roughly which fits the old Tanser model) with very few cases during the dry season. See the work by I. Diouf for example (<https://doi.org/10.3390/ijerph14101119>). Year-round transmission values in the Nile region must be backed up by medical papers / epidemiological studies as well. This overestimation appears clearly on Fig 2e-f where LisFlood predicts a very large surface suitable for 9-12 months. DRC values should be larger I think, this is one of the most endemic country in Africa. Basically, the authors should provide more details on these issues in discussion.

We have added extensive documentation of LIS-MAL performance over the Sahelian fringe and other margins of transmission suitability, incorporating observations of over twenty field studies. See Supplementary information S2 for an extensive treatment of this. We compared LIS-MAL and LIS-MAL + IRRRA with an indicative rainfall-thermal model calculated from the methods of Tanser qualitatively against published observations and descriptions of malaria and vector distribution, for four regions: The Sahel, Congo, Ethiopia and the Horn of Africa, and Southern Africa (new Figures S18 – S21). As part of this, we specifically report the comparison in Sénégal, the Nile and the DRC as identified by the review.

(3) Climate Model bias-correction?

A few points here. Firstly, the authors should emphasize that they are using a Regional Climate Model driven by an ensemble of 7 GCMs in the abstract. Secondly, the authors should use a bias-calibration technique. RCM driven by GCM can show large temperature and rainfall biases over Africa. Consequently, this point needs to be addressed or at least mentioned as an important caveat in discussion.

We added this to the Abstract. The relevant sentence now reads: *“To address this problem we use downscaled daily climate predictions from seven GCMs to run a continental-scale hydrological model for a process-based representation of mosquito breeding habitat availability.”*

We have added the following sentences to the relevant section of the Methods to state explicitly that bias correction was not performed, explain the reasons for this decision and acknowledge the resulting limitations: *“No bias correction was performed on the climate projections as it often breaks the physical coherence between the atmospheric variables, while its benefits in regions with sparse observational datasets as in this case are yet disputed (Muerth et al., 2013). Resulting runoff simulations were shown to have variable quality, with positive bias in arid regions, yet skilful in reproducing their sub-seasonal variability (Hirpa et al., 2019).”*

We have also added these reference list:

Muerth, M. J., Gauvin St-Denis, B., Ricard, S., Velázquez, J. A., Schmid, J., Minville, M., Caya, D., Chaumont, D., Ludwig, R. and Turcotte, R.: On the need for bias correction in regional climate scenarios to assess climate change impacts on river runoff, *Hydrology and Earth System Sciences*, 17(3), 1189–1204, doi:10.5194/hess-17-1189-2013, 2013.

Hirpa, F. A., Alfieri, L., Lees, T., Peng, J., Dyer, E. and Dadson, S. J.: Streamflow response to climate change in the Greater Horn of Africa, *Climatic Change*, doi:10.1007/s10584-019-02547-x, 2019.

(4) Malaria model description to be improved.

First, I think the authors should find an acronym for their malaria model (as Lisflood is already a hydrological model?). The main model description in methods should be extended in Supp Materials. A diagram or schematic showing the different input layers (irrigation, population etc.) and if they vary in space and/or time would be a good addition, or this information can also be included in a summary table.

In response to this comment we have now named our coupled model: LIS-MAL. We have edited this throughout the manuscript, Figures and Supplementary Information, reserving “Lisflood” for instances where we refer specifically to the Lisflood component of the broader workflow.

We have also added a new Figure (now Figure S2) to help with the model description and refer to this twice in the main text. As suggested, we explicitly detail the variability of each input layer of process step in both time and space.

Figure S2. LIS-MAL model structure. The spatial resolution of each input of calculation is given in the bottom right corner of each box, while the temporal variability is indicated along the left side.

(5) Other hydrological malaria models could be mentioned in Introduction

I am thinking about the work by Arne Bomblies & Yamana et al. These other models developed for small scale applications should be compared to the main author’s findings as well.

We have added reference to Bomblies et al. (2008) and Yamana et al. (2016) where the importance of water availability is first mentioned (detailed in the specific point below). Moreover, we added an additional sentence at the end of that paragraph, detailing the valuable contribution of this village-scale model and noting that this important coupling of hydrology and biology has yet to be undertaken at the continental scale: *“While detailed dynamic process-based models coupling hydrological and biological components of malaria transmission have been developed at the village-scale (Bomblies et al., 2008; Yamana et al., 2016), such hydrological representations are lacking in existing malaria climatic suitability estimates at the continental scale.”*

Finally, we also make comparison to the work of Yamana et al. (2016) in the discussion, as this work also highlighted a projected decrease in hydro-climatic suitability in West Africa.

Bomblies, A., Duchemin, J.B. and Eltahir, E.A., 2008. Hydrology of malaria: Model development and application to a Sahelian village. *Water Resources Research*, 44(12).

Yamana, T.K., Bomblies, A. and Eltahir, E.A., 2016. Climate change unlikely to increase malaria burden in West Africa. *Nature Climate Change*, 6(11), pp.1009-1013.

Minor points:

Intro: “...malaria deaths were reported in Africa.”

Change made as suggested

Intro: “... biting rates and individual longevity (add references)”

We added reference to the relatively recent paper of Shapiro et al. (2017) Quantifying the effects of temperature on mosquito and parasite traits that determine the transmission potential of human malaria. PLoS biology, 15(10), p.e2003489, which covers these topics.

Intro: “of both parasite and vectors”. Importantly the authors should investigate the recent work by Waite et al (<https://doi.org/10.1098/rsbl.2019.0275>) who showed lower temperature thresholds for sporogony in key Anopheles vectors.

We have added reference to this interesting paper at this point in the text.

Intro: “The availability ... of malaria transmission (add references to Bomblies and Yamana work)”

We have added reference to the following papers:

Bomblies, A., Duchemin, J.B. and Eltahir, E.A., 2008. Hydrology of malaria: Model development and application to a Sahelian village. *Water Resources Research*, 44(12).

Yamana, T.K., Bomblies, A. and Eltahir, E.A., 2016. Climate change unlikely to increase malaria burden in West Africa. *Nature Climate Change*, 6(11), pp.1009-1013.

Intro: “Moreover... not included in such models”. The authors could add a couple of sentences about potential dam effects on malaria, it’s a very important issue (see work by Solomon Kibret - <https://malariajournal.biomedcentral.com/articles/10.1186/s12936-015-0873-2>).

We have edited the sentence to add reference to Kibret et al. (2015) and mention explicitly reservoirs arising from the construction of large dams.

Page 7: “Lisflood projected percentage ... Tanzania”. This sentence is poorly written, please reword

We have restructured this sentence to read: “Percentage increases in population-months estimated by LIS-MAL are more focused around east Africa, particularly Somalia and Tanzania.”

Conclusion: “... could be attributed to climate alone”. The work by Gething et al. should also be cited here (<https://www.nature.com/articles/nature09098>)

This paper is now cited here.

REVIEWER 2

This is a well-written paper that develops the science on malaria hazard. The authors show that the use of a hydrological model to map climatic suitability for malaria transmission produces a better match with the pre-intervention malaria map of Lysenko and Semashko. This is a nice result and provides confidence in their subsequent predictions of future climatic suitability.

We would like to take the opportunity to thank the reviewer for these positive comments regarding the paper.

I have been asked to comment on the hydrological modelling specifically. I believe LISFLOOD is a good choice of model for this application, and my only slight query is with the use of the 30m³s⁻¹ minimum threshold for suitability. In particular, the authors state that the area estimated to be hydro-climatically suitable for stable malaria transmission is smaller than previous models suggest. I wonder if this result stands if a different threshold is used.

Firstly, for clarity, I would recommend that the authors cite the source of this threshold (Kiszewski et al. , 2004) within the main body of text (rather than just in the supplementary information). Secondly, I would like to see a sensitivity analysis of this threshold (not necessarily in the main body of text, but definitely in the supplementary material). There is always a challenge in relating the real world to the model world, especially to models with a coarse spatial representation of hydrology. The suitability

threshold should be seen as an effective parameter, in that it is used to attempt to represent suitable accumulations of water far smaller than the grid scale. Is the area still smaller than previous models suggest if different values for this suitability threshold are used?

In response to these comments we undertook a further sixty model runs and have added an item of Supplementary Information (S1. Sensitivity analysis of model parameters) that details a sensitivity analysis of both the runoff threshold of LIS-MAL and the rainfall threshold of previous models. See also new figures S14-S16 to support this additional item.

LIS-MAL suitability estimates were not sensitive to the runoff threshold within a 50% range. Decreasing the runoff threshold by 50% increased the suitable area by 10%, while raising the threshold by 50% decreased the suitable area by just 6%. The suitability estimates were more sensitive at lower runoff thresholds ($<10 \text{ m}^3 \text{ s}^{-1}$) and more sensitivity was observed in the 3-month categories of season length. Further details are provided in Supplementary Information S1.

We have added a sentence to the main text to direct readers to this Supplementary Information and present this headline finding: *“A sensitivity analysis for this threshold is presented in Supplementary Information S1: overall, LIS-MAL suitability estimates were not sensitive to the runoff threshold for values \pm 50% (increasing and decreasing the threshold value by 50% changed the estimated suitable area by -6% and 10% respectively).”*

REVIEWER 3

This study applies a continental-scale hydrologic model combined with malaria temperature suitability to produce a mapping of current and future climate suitability for malaria transmission. In theory, the addition of hydrological processes and irrigation should be an improvement over models that use rainfall alone as they would provide a more accurate measure of suitable breeding habitat for the Anopheles mosquitoes that transmit malaria.

However, there is little validation of the model to prove that this is the case. Understandably – validation is challenging, as the authors note. The validation presented here consists of a visual comparison to the map of malaria transmission limits in 1900 constructed by Lysenko & Semashko by interpolating historical records with climate data and expert opinion. From Figure 2 of the paper, it is not obvious to me that the new model outperforms the Tanser model at reproducing the Lysenko map. From Figure S4, the model appears to do rather poorly at the northern and southern limits of malaria transmission, as well as in the East African highlands. This discrepancy is problematic, as these are precisely the areas that are most sensitive to changes in transmission potential due to climate change.

See comments to specific points as listed below. We agree that: (i) validation is challenging; and (ii) the new model does not especially outperform the Tanser model. The quantitative validation we have added in Supplementary information S2 suggests that there is little difference between moderate rainfall threshold values, precipitation to evaporation ratio of Lindsey and the LIS-MAL model. We have also detailed comparisons between models and observed data at the limits of malaria transmission in this Supplementary Information which indicates that the LIS-MAL model is much better at representing the complex transmission patterns observed at these fringes.

Interestingly, the hydrological model presented here is also applied as a simple threshold rule ($30 \text{ m}^3/\text{s}$ over a 0.5-degree grid cell). It is unclear how this threshold relates to the fine-scale hydrological processes that determine anopheles habitat availability, given the species' preference for puddles and small water pools on the order of meters to tens of meters.

The runoff threshold we applied is grid-scale dependent and represents a 1 mm depth of runoff averaged over the grid cell per day. In future, we hope to reduce the grid size used in such models and have already

investigated the use of hydro-dynamic models for this purpose which could reduce this grid size substantially. However, as noted in response to Reviewer 2, LIS-MAL suitability estimates were not sensitive to the runoff threshold within a 50% range. Decreasing the runoff threshold by 50% increased the suitable area by 10%, while raising the threshold by 50% decreased the suitable area by just 6%. See the new Supplementary Information S1 for further details.

While the inclusion of hydrology and irrigation are useful and there are some interesting results presented as far as climate change projections, I believe the model must first be more convincing in defining malaria suitability in the present climate before we can begin to consider its ability to make projections in a new climate.

Please see Supplementary Information S2 for a detailed comparison of model output with observations, with a particular focus at the fringes of modelled and observed suitability.

Specific comments:

- (1) The model needs to be better validated. Comparison with the Lysenko map should be better quantified. Additional comparisons to higher quality data, for example malaria prevalence surveys such as those use by Tanser or anopheles occurrence surveys, would greatly improve the credibility of the model. The Malaria Atlas Project provides some such data on their website.**

We have added a Supplementary Information item on validation against observations. We now validate our model against four datasets: (1) Lysenko and Semashko (1968) pre-intervention (~1900) map of malaria endemicity in Africa; (2) Kyalo et al. (2017) anopheles inventory of sub-Saharan Africa (1898-2016) observations of anophelines. Furthermore, we investigated (and indeed performed the validation) using (3) Malaria Atlas Project modelled estimates of parasite rate from Bhatt et al. (2015) for 2015 and (4) Wiebe et al. (2017) estimate of the mean modelled relative probability of occurrence of *Anopheles gambiae*; however, note the comment above that these latter data sets rely in some part on gridded climate data for interpolation and cannot therefore serve for validation analysis independent of climate. For further details, see comments in response to Reviewer 1.

- (2) The discrepancy at the northern and southern limits of climate suitability, as well as East Africa, should be addressed.**

As noted above, the new Supplementary Information S2 provides a detailed comparison of model results with observations at the limits of climate suitability.

- (3) The choice of the rainfall runoff threshold value should be discussed, and sensitivity to this value should be assessed.**

See comments of Reviewer 2 above. We have added an item of Supplementary Information (S1. Sensitivity analysis of model parameters) that details a sensitivity analysis of this threshold.

- (4) The discussion should mention the disconnect between the scale of 55km grid cell hydrologic modeling and finer-scale processes determining availability of anopheles breeding habitat.**

We have expanded our discussion of this in the Conclusions section which now reads

“This includes the potential for future high resolution models: to quantify water body dynamics at catchment and floodplain scales a hydrodynamic river routing model³⁰ could be used to represent flood expansion and contraction at sub-km spatial resolutions. This would also allow a more complex environmental estimate of malaria hydro-climatic suitability by enabling the modelling of finer-scale hydrological processes determining the availability of anopheles breeding habitat, permitting other

physical processes (e.g. larval flushing effects of high velocity river flows) to be included and for vector niches to be represented explicitly.”

Certainly, the reduction of the grid size is a focus for further research for this exact reason.

- (5) There should be mention of the work in the decade+ following Tanser and other simple threshold models that use hydrologic processes and/or mosquito larval biology in determining climate suitability for malaria**

Please see the response to Reviewer 1 for which we have added reference to a number of such studies.

- (6) This is more a matter of preference, but I believe there is too much emphasis in the manuscript on drawing comparisons between projections under Tanser’s formulation vs Lisflood – I don’t believe the Tanser projections are considered to be definitive or a consensus view. If a comparison to other maps or projections is necessary, there have been a number of more recent projections that may be more relevant**

We understand this point; however, a comparison of all ten hydro-climatic models throughout the main text of the manuscript was somewhat unmanageable. Our intention is more to compare the use of a rainfall threshold in the representation of water bodies with a hydrological model and highlight the different patterns of suitability. We present a comparison with all ten hydro-climatic models in the Supplementary Information, as highlighted in the following text:

“Herein we primarily compare the LIS-MAL hydrological model estimates with a moderate rainfall threshold of 60 mm per month alongside single catalyst month of 80 mm rainfall as applied by Tanser et al.⁷; comparison with other published rainfall thresholds is presented in the Supplementary Information.”

The selection of the Tanser threshold was made as it is a moderate threshold (i.e. not an extreme value from the literature) that performs relatively well in the validation and is a threshold value that has been applied extensively (e.g. MARA).

Reviewers' Comments:

Reviewer #1:

None

Reviewer #2:

Remarks to the Author:

The authors have revised the paper to include a sensitivity analysis as I suggested, this analysis is well described and benefits from the inclusion of two figures (S13 and S14). I would also add that the inclusion of a detailed evaluation of the model in the supplementary material is a valuable addition to the paper. I'm happy to recommend this for publication in Nature Communications.

Best wishes,
Liz Stephens

Reviewer #3:

Remarks to the Author:

The authors have thoroughly addressed each of my comments. I am happy to recommend the publication of this manuscript.